# 3D Digital Modeling as a Sustainable Conservation and Revitalization Path for the Cultural Heritage of Han Dynasty Stone Reliefs

Difei Zhao [1,2,*] , Chaowei Liu [3], Xinyue Zhang [4], Xiaoyue Zhai [5], Yinglan Deng [6], Hongyu Chen [7], Juju Hu [1], Dandan Liu [4] and Pingjia Luo [6,*]

1. Artificial Intelligence Research Institute, China University of Mining and Technology, Xuzhou 221000, China; ts14160129@cumt.edu.cn
2. International College, Krirk University, Bangkok 10220, Thailand
3. School of Humanities and Arts, China University of Mining and Technology, Xuzhou 221116, China; 18203720@cumt.edu.cn
4. Sunyueqi Honors College, China University of Mining and Technology, Xuzhou 221116, China; 22215736@cumt.edu.cn (X.Z.); 08212835@cumt.edu.cn (D.L.)
5. Xuzhou Han Dynasty Terracotta Warriors Museum, Xuzhou 221116, China; 798953878f@gmail.com
6. School of Architecture and Design, China University of Mining and Technology, Xuzhou 221000, China; 09183162@cumt.edu.cn
7. Faculty of Architecture and City Planning, Kunming University of Science and Technology, Kunming 650500, China; 385603495@kmu.edu.cn
* Correspondence: difei.zhao@cumt.edu.cn (D.Z.); luopingjia@cumt.edu.cn (P.L.)

**Abstract:** Cultural relics and historical sites serve as carriers of cultural, historical, and artistic information. However, any damage incurred by these cultural relics can result in the loss of information, consequently impacting sustainable conservation and revitalization of the cultural heritage. Han Dynasty stone reliefs are a representative carrier of art and history during the Han Dynasty, an early stage of Chinese history. Due to the influence of materials, carving techniques, and protective measures, the conservation and revitalization of Han Dynasty stone reliefs have been significantly restricted. In this study, a systematic investigation was carried out to study the current situation and existing problems related to the protection of Han Dynasty stone reliefs. Additionally, a case study was conducted using the Wuling Ancestral Hall (Wuliang Shrine) as an example, to explore the integration of 3D digital technology as a new sustainable approach. The results show that natural weathering and conventional techniques have caused irreversible information loss. Thus, adopting a three-dimensional digital perspective is crucial when considering the information preservation and revitalization of Han Dynasty stone reliefs. To achieve this, 3D digital models of representative stone reliefs, tomb chambers, and other sculptures from the Wuliang Ancestral Hall were established. These models provide new paths for accurately recording 3D information and better utilizing cultural heritage. Faced with the challenge of preserving historical heritage and its associated information, a workflow including 3D scanning, data collection and processing, 3D modeling, visualization, and information utilization is proposed. This approach offers new approaches for sustainable conservation and revitalization of Han Dynasty stone reliefs.

**Keywords:** Han Dynasty stone reliefs; sustainable preservation; cultural heritage; digital conservation; interdisciplinary research

## 1. Introduction

Han Dynasty stone reliefs are architectural stones carved with images in Han Dynasty architecture [1–4]. These stones are predominantly used in underground tombs, burial chambers, cemetery temples, gate towers of tombs, and gate towers of shrines. Their main function is as integral elements of the funeral ritual architecture, making these stone reliefs

essentially artistic works of sacrificial burial during the Chinese Han Dynasty period [4,5]. The artistic techniques and stylistic characteristics employed in these carved artworks signify a significant milestone in the evolution of classical Chinese art development. They exemplify the apex of classical Chinese art during the Han Dynasty and earlier periods, exerting a lasting influence on subsequent art. They hold a significant position in the history of Chinese art, bridging the past and the future [4–7]. Furthermore, these stone reliefs contain a diverse array of subjects, spanning from scenes of daily life, and historical events to representations of classical literary works, societal structures, local customs, mythological legends, and mythical beasts [4]. As crucial historical artifacts, they showcase the diverse social life of the Han Dynasty, serving as important mediums for documenting Chinese history and possessing high archaeological value [8,9].

As important historical relics, the Han Dynasty stone reliefs carry significant artistic value and historical and cultural information. Currently, the protection of Han Dynasty stone reliefs mainly involves their transfer to museums for preservation, while a small portion undergo on-site protection near their original locations. Some Han Dynasty stone reliefs have had their surface images replicated through rubbing. However, the Han Dynasty stone reliefs primarily utilize limestone as their building material, which, although relatively hard, is susceptible to weathering and human-induced damage. Surface weathering or damage to Han Dynasty stone reliefs can result in the permanent loss of information and the loss of important historical and cultural information. At present, the protection of Han Dynasty stone reliefs primarily relies on centralized collection and protection by museums and archaeological research institutions. Only a small portion of the original sites receive protection. However, there appears to be a lack of attention toward addressing surface weathering, damage, and the continuous loss of information. In the protection of information on stone reliefs, rubbing and imaging are the main methods employed. However, the rubbing method can cause additional surface damage, and it is difficult to preserve the three-dimensional structural information of the surface of stone reliefs through imaging. These conventional protection methods have become increasingly difficult to adapt to higher protection requirements and fail to provide a digital information foundation for activating cultural heritage. Presently, rapid developments in digital technology and intelligent technology have become important new approaches to cultural heritage preservation [10–14]. Han Dynasty stone reliefs combine painting and sculpture arts [4]. Meanwhile, it is also a combination of tangible and intangible heritage [12]. The rapid development of digital technology provides a new technical pathway for the protection and revitalization of cultural heritage such as Han Dynasty stone reliefs.

Xuzhou held a special status during the Han Dynasty. It was the hometown of the first emperor of the dynasty, Liu Bang, and the political center of the feudal state of the Han Dynasty [15]. There are numerous cultural relics such as Han pictorial stone tombs, ancestral halls, and other cultural relics, making Xuzhou an important origin of the production of Han Dynasty stone reliefs art. Xuzhou and its surrounding areas is a concentrated region for the preservation of Han Dynasty stone reliefs. In recent years, the Xuzhou area has witnessed continuous discoveries of Han Dynasty stone reliefs, presenting both new opportunities for cultural and historical research and fresh challenges concerning effective sustainable conservation and revitalization. Although museums and other institutions have effectively protected cultural relics, issues such as surface damage and information loss persist. Conventional methods of cultural relic protection are no longer sufficient to support the revitalization of these cultural relics. In addition, the utilization of cultural heritage information to foster cultural innovation in the future demands further research. To address these concerns, a comprehensive investigation of the current situation is essential. This will lay the groundwork for establishing a more systematic work program and workflow based on advanced technological methods. This approach aims to achieve three-dimensional digital conservation and revitalization of the cultural heritage of Han Dynasty stone reliefs. Through the preliminary investigate on the conservation and revitalization of Han Dynasty stone reliefs in the Xuzhou area, the

research team found that the lack of effective innovative technological paths for information preservation and utilization has significantly restricted the sustainable protection and intangible resource utilization of these cultural heritage. Through field surveys, field investigations, expert interviews, and experimental research on 3D modeling, this study analyses the problems in the conservation of Han Dynasty stone reliefs. Taking Xuzhou as an example, this study reveals the significance and urgency of the digital conservation of Han Dynasty stone reliefs. Based on this, a workflow for the conservation of Han Dynasty stone relief cultural heritage based on 3D digital technology is proposed, providing scientific guidance for the scientific protection of Han Dynasty stone relief cultural heritage. The main purposes of this study are as follows: (1) to investigate the conservation of Han Dynasty stone reliefs, a special cultural heritage from the perspective of petrology and geology; (2) to investigate how to use 3D digital technology to record, preserve, and utilize information of Han Dynasty stone reliefs in conjunction with case studies; (3) to summarize a workflow to guide the sustainable digital preservation of Han Dynasty stone reliefs in the region.

## 2. Study Area and Methodology

Xuzhou and the surrounding areas were selected to carry out research on the conservation of Han Dynasty stone reliefs. The research team conducted research and fieldwork in Xuzhou, Jining, Tengzhou, Nanyang, Lianyungang, Linyi, Huaibei, Suzhou, Zibo, and other locations. The survey and fieldwork sites are shown in Figure 1. These sites are mainly located in four provinces: Jiangsu, Shandong, Anhui, and Henan, which are the main areas where Han Dynasty stone reliefs are unearthed or preserved in East China. The research team joined forces with researchers from local government cultural and conservation institutions to carry out the surveys and interviews. Xuzhou, Nanyang, and the southern part of Shandong Province were selected as areas with a high concentration of Han Dynasty stone reliefs.

An interdisciplinary research approach was adopted to explore the research objectives. The research team brought together researchers from the fields of geology (petrology), history, informatics, and architecture to investigate the conservation and preservation of Han Dynasty stone reliefs in the region. On this basis, the research team further engaged researchers from the fields of artificial intelligence and computing to design conservation solutions and workflows to address the problems revealed. The research used a combination of field surveys, fieldwork, expert interviews, experimental studies, and case studies. Firstly, the team conducted systematic fieldwork and surveys work in the research area to gain a comprehensive understanding of the current situation and technical methods used in protecting Han Dynasty stone reliefs. During this process, the research team engaged in extensive communication with professional technical personnel from museums and related institutions, obtaining data and materials. On this basis, the research team selected the Wuliang Ancestral Hall (Wuliang Shrine) in Jining, Shandong Province as an example to conduct a case study. In the case study, the research team not only conducted research on the conservation status of Han stone reliefs and other cultural heritage in the Wuliang Ancestral Hall, but also attempted digital conservation of cultural heritage with different types and preservation status through technical methods such as 3D scanning, 3D modeling, and 3D printing. Finally, the research team combined the results of field research, expert interviews, 3D scanning, 3D modeling, and 3D printing to form a new workflow to better guide sustainable digital conservation and revitalization of Han Dynasty stone reliefs. The logic of the methodology is to identify existing problems and research gaps, then designing a new workflow through a case study to private potential new solutions for sustainable conservation and revitalization of Han Dynasty stone reliefs. In the methodology, this study innovatively applied the research process of status investigation, problem identification, case study, and experience summary. Meanwhile, due to the unique research objectives, this study adopts an interdisciplinary approach to conduct research.

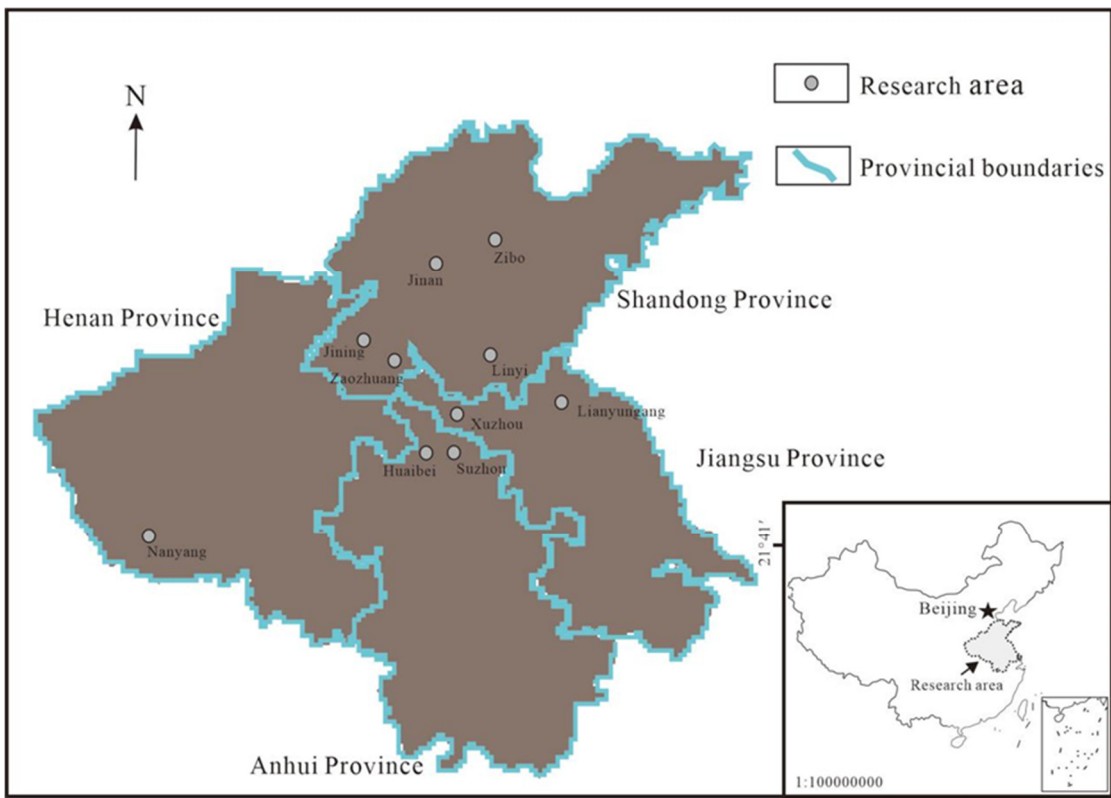

**Figure 1.** Location of research area.

## 3. Conservation of Han Dynasty Stone Reliefs

Han Dynasty stone reliefs are the largest collection and the most content-rich Han Dynasty artworks. These stones are a form of carved art and also form part of the architecture of tombs or ancestral halls. The unearthed or preserved Han Dynasty stone reliefs in the study area are mainly composed of limestone. The carving techniques of Han Dynasty stone reliefs can be divided into several categories, including single-line shading, reduced-ground flat carving, shading line carving, reduced-ground relief carving, and round carving. Some Han Dynasty stone reliefs combine different carving techniques, such as reduced-ground carving with shaded lines, and the use of shaded lines on flat protruding figures to express the details. Therefore, although Han Dynasty stone reliefs are carvings, their main information is focused on the surface, emphasizing the use of lines. Some Han Dynasty stone reliefs in the Nanyang area use the "boneless" method of traditional Chinese painting, organically merging with relief art. The edges of the relief are accentuated by lines, enhancing the three-dimensionality of the images, while the use of lines creates a sense of depth in the composition. This combination of painting and carving techniques enhances the expressive power of the artworks. However, as the focus is mainly on the surface and shallow lines and carvings of the rocks, they are susceptible to weathering and other damaging factors. Note that as carved artworks, Han Dynasty stone reliefs contain two-dimensional and three-dimensional information, which to some extent increases the difficulty of comprehensive protection. Scholars have also pointed out that in understanding and interpreting Han dynamic bricks and stone empirical reliefs, observers should adopt a three-dimensional viewpoint to describe the motifs [8].

During the investigation of the primary institutions and sites for preserving Han Dynasty stone reliefs in the study area, the research team found that there are a number of protection issues. These include surface damage caused by weathering and human-induced damage (such as mechanical damage or damage from rubbings) (Figure 2). Among these, the surface weathering of Han Dynasty stone reliefs is a more serious problem, which is related to their chert material. Some of the Han Dynasty stone reliefs had to be preserved

in the open due to space constraints (Figure 2a), which accelerated the weathering process on their surfaces, resulting in the rapid loss of important information (Figure 2b,c). Some of the Han Dynasty stone reliefs were not protected in a timely and centralized manner. Some of the Han Dynasty stone reliefs have also been damaged by mechanical fractures or other human factors (Figure 2d,e). In particular, Han Dynasty stone reliefs carved with incised lines are more susceptible to information loss due to their shallow surface depth.

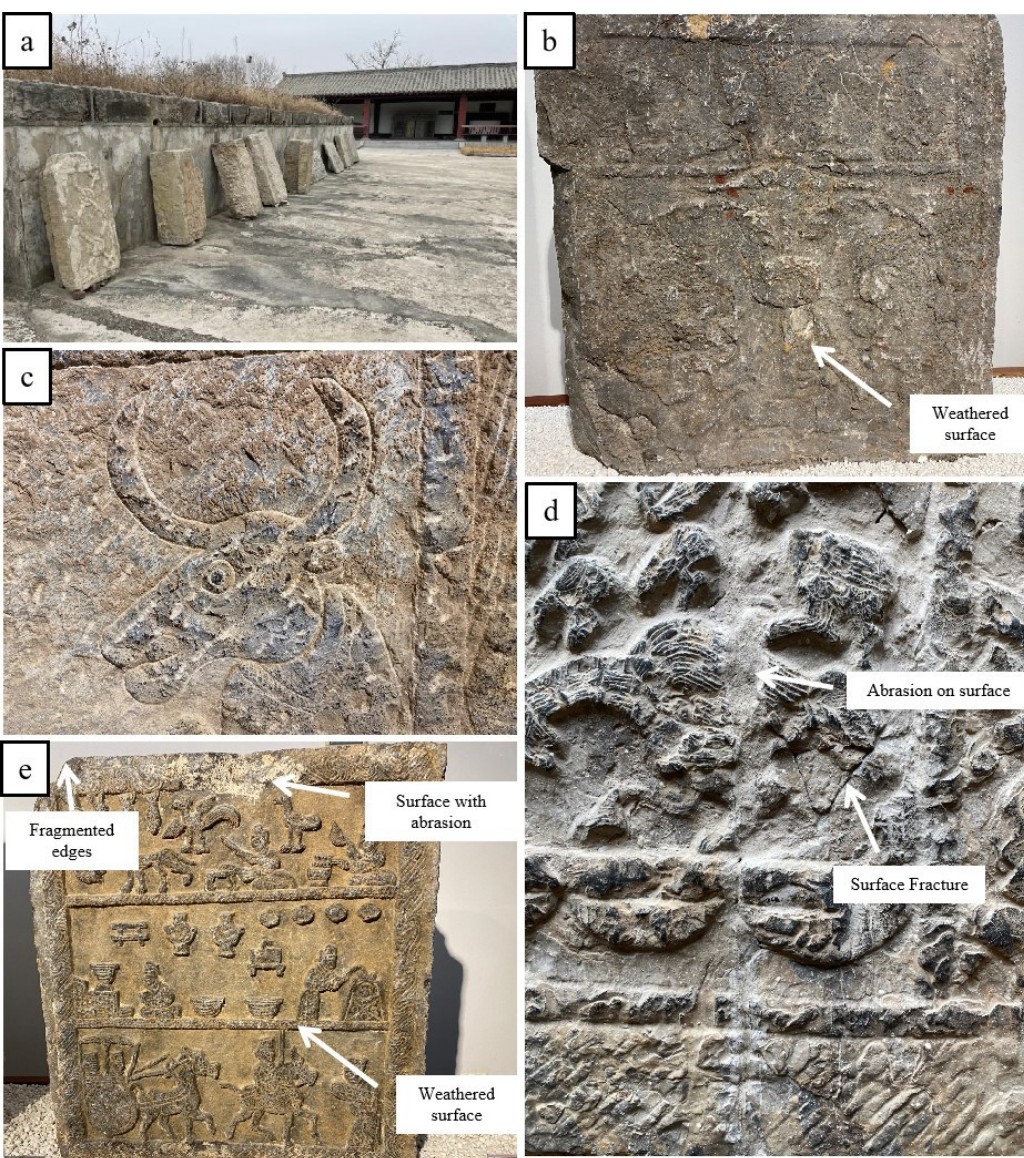

**Figure 2.** The conservation situation and existing problems of Han Dynasty stone reliefs obtained through field research and field research. (**a**) Han Dynasty stone reliefs preserved in the open; (**b**,**c**) information loss due to weathering; (**d**,**e**) Han Dynasty stone reliefs damaged by mechanical fractures or other human factors.

In the past, due to technical constraints, the replication and dissemination of surface information on Han Dynasty stone reliefs relied mainly on rubbings. Rubbings have long-term effects on the surface of the stone reliefs. Firstly, the surface of pounded stones is usually left with ink stains that are difficult to remove. As the ink residue is corrosive, it can be left on the surface of rocks and carvings for an extended period, damaging the original status of the cultural relics, especially the small lines or ornaments. There is also mechanical damage caused by human manipulation in the process of topiary. Currently, in order to better protect cultural relics, China no longer permits rubbings of the surface of

Han Dynasty stone reliefs for the better conservation of cultural relics, but this also brings new challenges in how to effectively extract, preserve and disseminate cultural and artistic information [16]. Additionally, rubbings can only preserve two-dimensional information, and their artistic level is lower compared to that of the original Han Dynasty stone reliefs, with its form but not its charm and its aesthetic effect is insufficient. It is not possible to gain an understanding of the different carving techniques from the topiary, nor does it show the hierarchy of carvings, and it may even mislead the understanding of the artistic qualities of Han Dynasty stone reliefs.

The conservation and study of Han Dynasty stone reliefs in China has a long history and can be divided into four stages based on their excavation and preservation processes (Figure 3). Prior to the Qing Dynasty, the focus was primarily on the collection and initial collation of fragmentary preserved Han painting stone materials and rubbings (epigraphy stage). Starting from the Song Dynasty, many epigraphers collected and accumulated a wealth of rubbings of Han Dynasty stone reliefs, and carried out preliminary examinations of the inscriptions on the inscriptions and the content of historical stories and portraits. From the late Qing Dynasty to the 1980s, archaeological and specialized research on the stone reliefs gradually unfolded, resulting in the scientific documentation and measurement of the stone reliefs and their contents. This development led to the inclusion of the Han pictorial stone research in the realm of formal archaeological science (archaeological collection stage). During this stage, the information on Han Dynasty stone reliefs gradually became more abundant and comprehensive, providing a better understanding of their carving techniques, content interpretation, regional distribution, and artistic characteristics. Since the 1980s, comprehensive studies were launched, and a more comprehensive understanding of the characteristics from different regions and periods was gained. In the new century, however, with the widespread of computer technology, digital technology, and the rapid development of intelligent technology, more cutting-edge technologies have the potential to be applied to the conservation of Han Dynasty stone reliefs. More importantly, these new technologies also offer new technical paths for the revitalization of cultural heritage on the basis of conservation. In recent years, there have been cases of cultural heritage preservation using technologies such as 3D modeling, 3D visualization, and virtual reality in the conservation of cultural heritage, such as the digital conservation of the Dunhuang Mogao Cave [17,18].

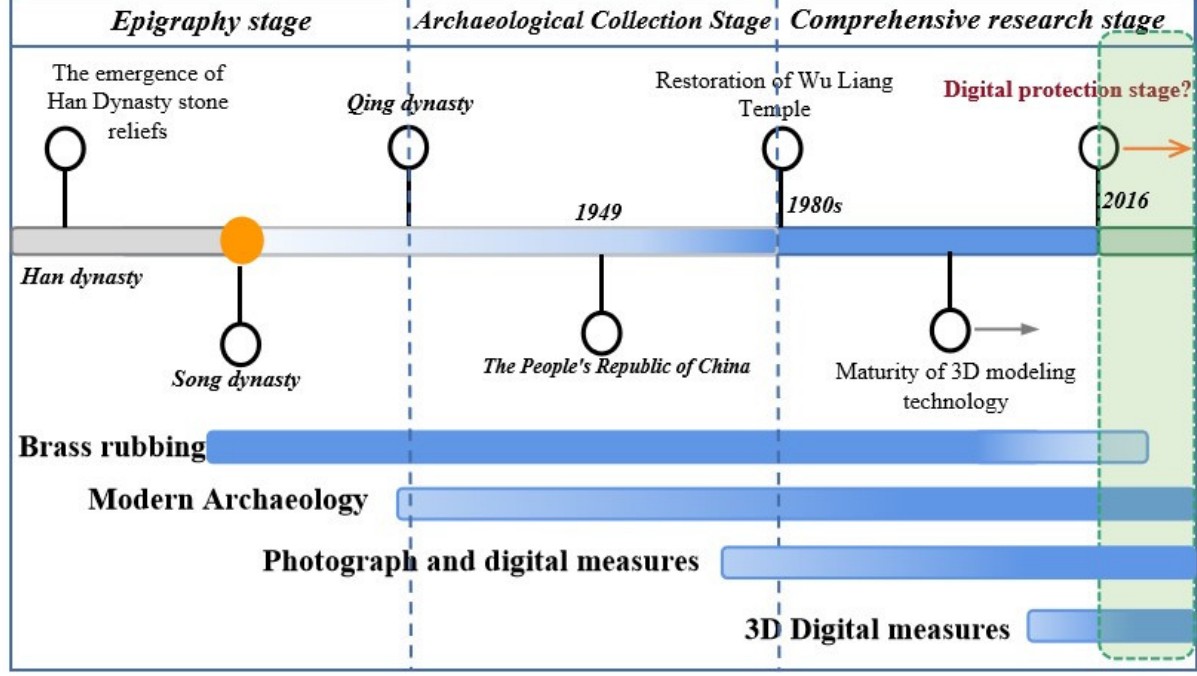

**Figure 3.** The four stages of conservation and research on Han Dynasty stone reliefs in China.

## 4. Three-Dimensional Digital Conservation Measures for Han Dynasty Stone Reliefs

After conducting research on the conservation of Han Dynasty stone reliefs in the study area, the research team came to the following conclusions: (1) the fine structural information of Han Dynasty stone reliefs is easily damaged by natural weathering or human factors due to their materials and surface carvings; (2) in the natural environment, unprotected Han Dynasty stone reliefs are destroyed by weathering very quickly and the loss of information is irreversible; (3) conventional methods of extracting information from Han Dynasty stone reliefs (such as rubbings) have a destructive effect, and the information obtained is limited to two-dimensional information, which does not reflect their structural details and value as carvings; (4) there are still relatively few cases of digital conservation of Han Dynasty stone reliefs using new technologies, and no standard workflow has been formed to guide the scientific conservation of Han Dynasty stone reliefs. Therefore, there is an urgent need for the digital conservation of Han Dynasty stone reliefs.

Thus, the research team carried out further research and case studies using the example of Han Dynasty stone reliefs and Han Dynasty carvings in and around the Wuliang Ancestral Hall (Wuliang Shrine) in Jining, Shandong Province (Figure 4). The Wuliang Ancestral Hall is situated at the northern foot of Wuzhai Mountain in Jiaxiang County, southwest Shandong Province. It is the only surviving temple from the 2nd century A.D. and, together with the stone queue, stone lions, stone monuments, and tombs in the mausoleum, forms a magnificent and spectacular stone carving [8,19,20]. Such large-scale architectural structures required substantial wealth during that period. Rarely, the stone queue and stone lions are still preserved in their original locations. It is the largest and best-preserved Han Dynasty stone relief in China. The study of Han Dynasty stone reliefs not only includes the two-dimensional and three-dimensional information carried by the stones, but also the overall study of the place where the stones were located and the surrounding natural and humanistic environment.

Han pictorial stone burials are part of the configuration of Han Dynasty mausoleums, which consist of stone quoins, stone beasts of the divine path, ancestral halls, tombstones, sealed soil, and burials. The Wushi Cemetery in Jiaxiang, Shandong, is a clear chronological and fully configured Han Dynasty family mausoleum. The Wushi ancestral hall is a hot spot for academic research, but the attention to the stone queue, the stone lion of the sacred path, and the burial have not been sufficient. Further research is needed to explore the relationship between the buildings and the overall layout of the whole mausoleum. The scenes of the mausoleum restored by technical means are not only beneficial to the study of the configuration, function, and symbolic meaning of the images but also to the study of Han Dynasty architecture, ritual thought and funeral rites.

At present, Wuliang Ancestral Hall has a pair of stone queues, a pair of stone lions, two stone monuments (Figure 4a), four groups of more than 40 stones of stone carved components of the ancestral hall, and a complete chamber structure of the tomb pit (Figure 4b) are preserved at the Wuliang Ancestral Hall. A museum of stone carvings of the Wushi tomb group was built at the original site, which collects some Han Dynasty stone reliefs from the surrounding areas. Although the museum provides protection for Han Dynasty stone reliefs, some of the museum's collection has not been given indoor preservation conditions due to site conditions and other constraints (Figure 4c,d). Additionally, the museum also lacks cutting-edge technical means for the protection and exhibition of Han Dynasty stone reliefs, and the extraction of the stone images has only been performed in the form of rubbings. The display of the stone reliefs has been mainly in the form of originals as well as reproductions of rubbings (Figure 4e,f).

Therefore, the research team selected the stone lions of the Wuliang Ancestral Hall, the tomb chamber, the representative Han Dynasty stone reliefs in the exhibition hall, and the outdoor open-air stone reliefs to carry out 3D modeling and to discuss the conservation value of 3D digital modeling for each of these artifacts in different status of preservation.

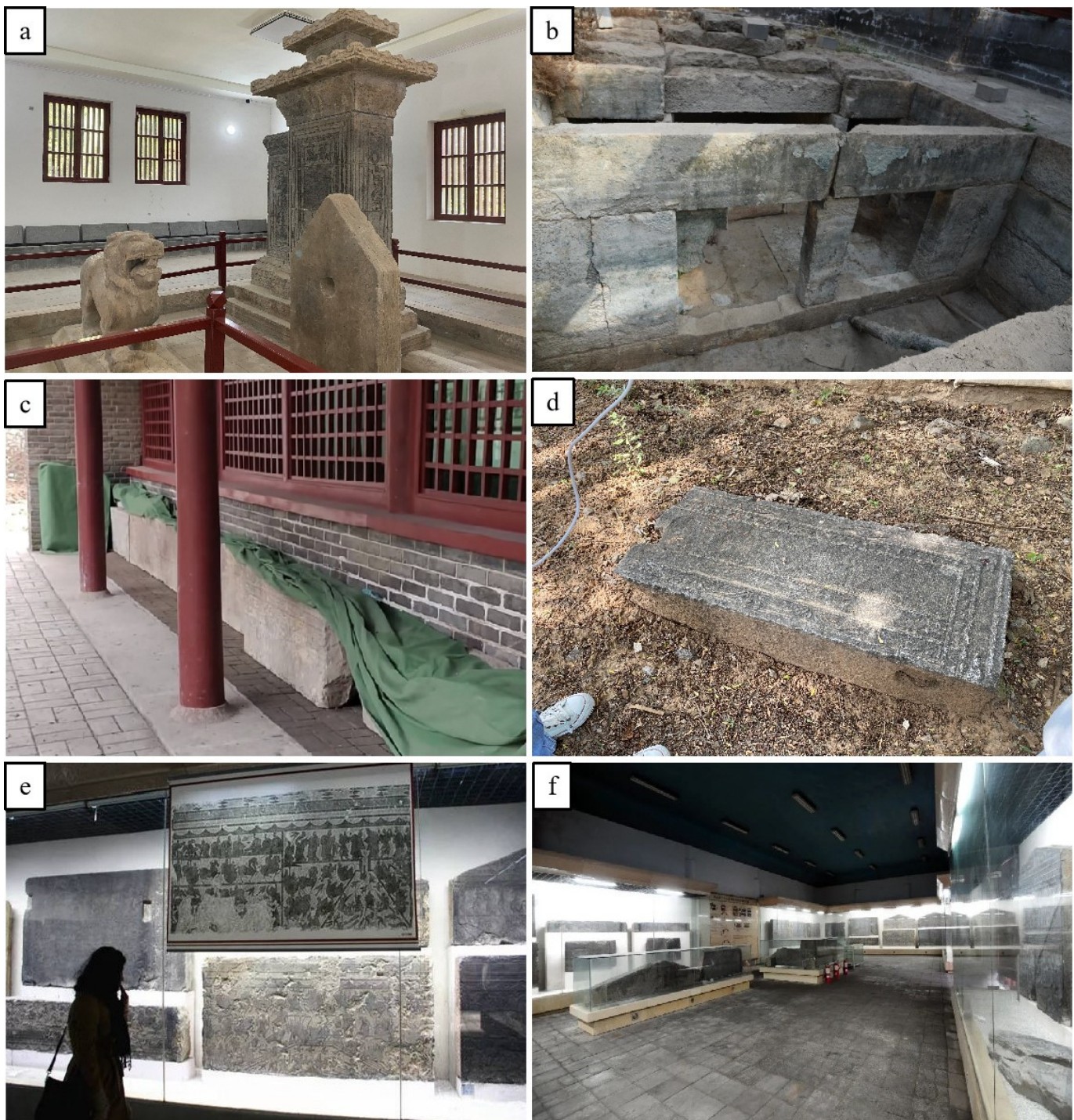

**Figure 4.** Research on the conservation measures of Han Dynasty stone reliefs for Wu Liang Temple. (**a**) Stone lion sculpture and stone tablet; (**b**) tomb chamber structure; (**c**) Han Dynasty stone reliefs protected outdoor with waterproof cloth; (**d**) Han Dynasty stone reliefs placed outdoors; (**e**) display method mainly based on original stone reliefs and rubbings; (**f**) exhibition hall interior.

### 4.1. Representative Han Dynasty Stone Reliefs in the Exhibition Hall

A representative indoor preserved Han Dynasty stone relief was selected for 3D modeling (Figure 5). Figure 5a shows a large area of slab-like stone reliefs. The relief is the east wall portrait of the Wuliang Ancestral Hall, which displays extremely rich contents, including mythological figures, historical stories, and life scenes such as the kitchen, carriage, and riding. The modeling results show that the model has a good ability

to record the details of ornamentation, images, and texts on the surface (Figure 5b), and they present the historical story of "Yu Rang assassinating King Xiangzi Zhao". The words "Yu Rang killed himself to avenge his confidant" are clearly recognized. At the same time, the point cloud data can be used to obtain detailed information about the relief and its surface image, such as depth, size, etc. The relief shown in Figure 5c has a more severely weathered surface and records 'The Carriages and Hoarsening on a Journey'. Figure 5d depicts a Han Dynasty stone relief preserved in a corner of a corridor, with a tarp to reduce the effects of weathering. The research team utilized 3D modeling technologies to complete the 3D data scanning and modeling of this relief in a relatively short time. The surface details can be clearly presented in the visualized model (Figure 5e). Since the weathering of the Han Dynasty stone reliefs preserved indoors is relatively slow, 3D modeling can provide new ways to enrich the display effect based on the preservation of their data, such as combining with virtual reality technology or 3D imaging technologies.

*4.2. Representative Han Dynasty Stone Reliefs Stored Outdoors*

3D Digital Modeling can be used to rapidly record the details of Han Dynasty stone reliefs that are scattered in the field or in situ and have not been further protected, preserving a digital version of the artifact. The research team selected outdoor Han Dynasty stone reliefs for 3D modeling, the preservation of which is shown in Figure 4d. The model created is shown in Figure 6. The model can both preserve the scene in which it is located (Figure 6a) and record its detailed information (Figure 6b). The advantages of 3D digital modeling in rapid documentation can play an important role in field surveys and timely digital conservation.

*4.3. Tomb Chamber Structure*

The Han Dynasty stone reliefs exhibit variations in form, size, and function depending on their different positions within the burial chambers. Therefore, recording the tomb chamber structure is of some significance for an in-depth understanding of Han Dynasty stone reliefs and helps to deepen the understanding of the funerary system during the Han Dynasty. The research team selected tomb no. 2 for the scenario modeling of the tomb chamber structure, as shown in Figure 7.

For a long time, there has been a lack of burial data for the Wu Family mausoleum in Jiaxiang, Shandong. In 1981, an archaeological team conducted an archaeological excavation of two burials in the southeast of the Wu Family mausoleum in Jiaxiang, Shandong. Both tombs faced northwest and were situated 68 m away from the stone queue, exactly in the same direction as the stone queue and the ancestral hall, and belonging to the burials in the Wu family mausoleum. Thus, the shape of a Han Dynasty family mausoleum gradually became clear (an image of the reconstructed cemetery can be inserted here). The entire mausoleum faced northwest, with a 6 m-wide divine path leading into it. At the end of the divine path stood a pair of double quays, each measuring 4.3 m in height, while 7 m in front of the quays were a pair of divine path stone lions. Behind the quays were five stone steles, followed by four ancestral halls, and behind the ancestral halls lay the family burial grounds. During the field research, an area in the eastern part of the mausoleum is still called the "Imperial Forest" by the locals, which is a small forest of pines and cypresses slightly higher than the surrounding land, or the remains of the Wu family cemetery. The complete and orderly layout of the buildings in the mausoleum reflects the ideological concept of the unity of heaven and man during the Han Dynasty.

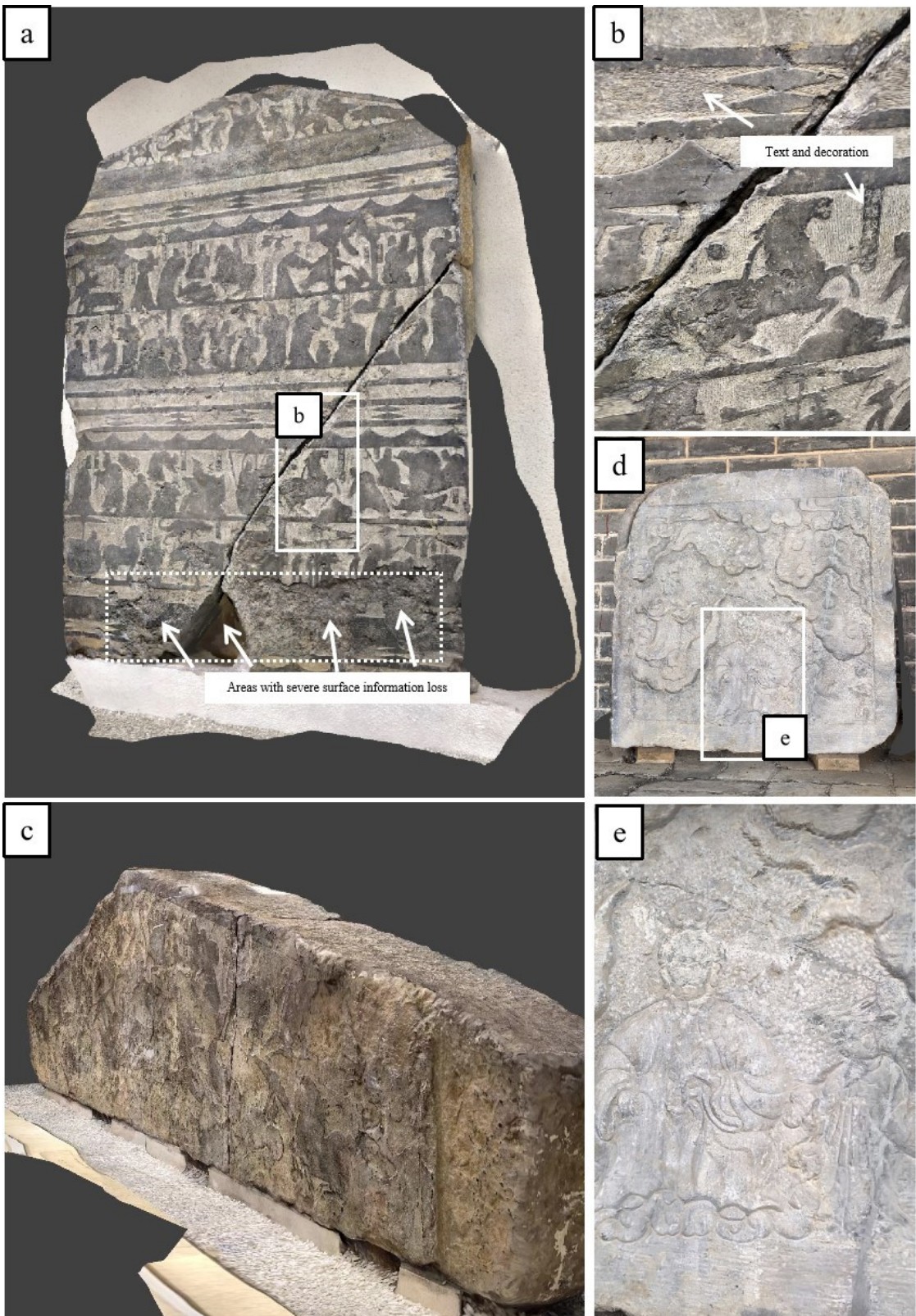

**Figure 5.** Three-dimensional digital modeling for representative Han Dynasty stone reliefs in the exhibition hall. (**a**,**b**) Three-dimensional model of a Han Dynasty stone relief with developed cracks. The lower surface has been damaged to a certain extent; (**c**) 3D model of a Han Dynasty stone relief that records 'The Carriages and Hoarsening on a Journey'; (**d**,**e**) 3D model and surface details of a Han Dynasty stone relief placed at a corner of the wall.

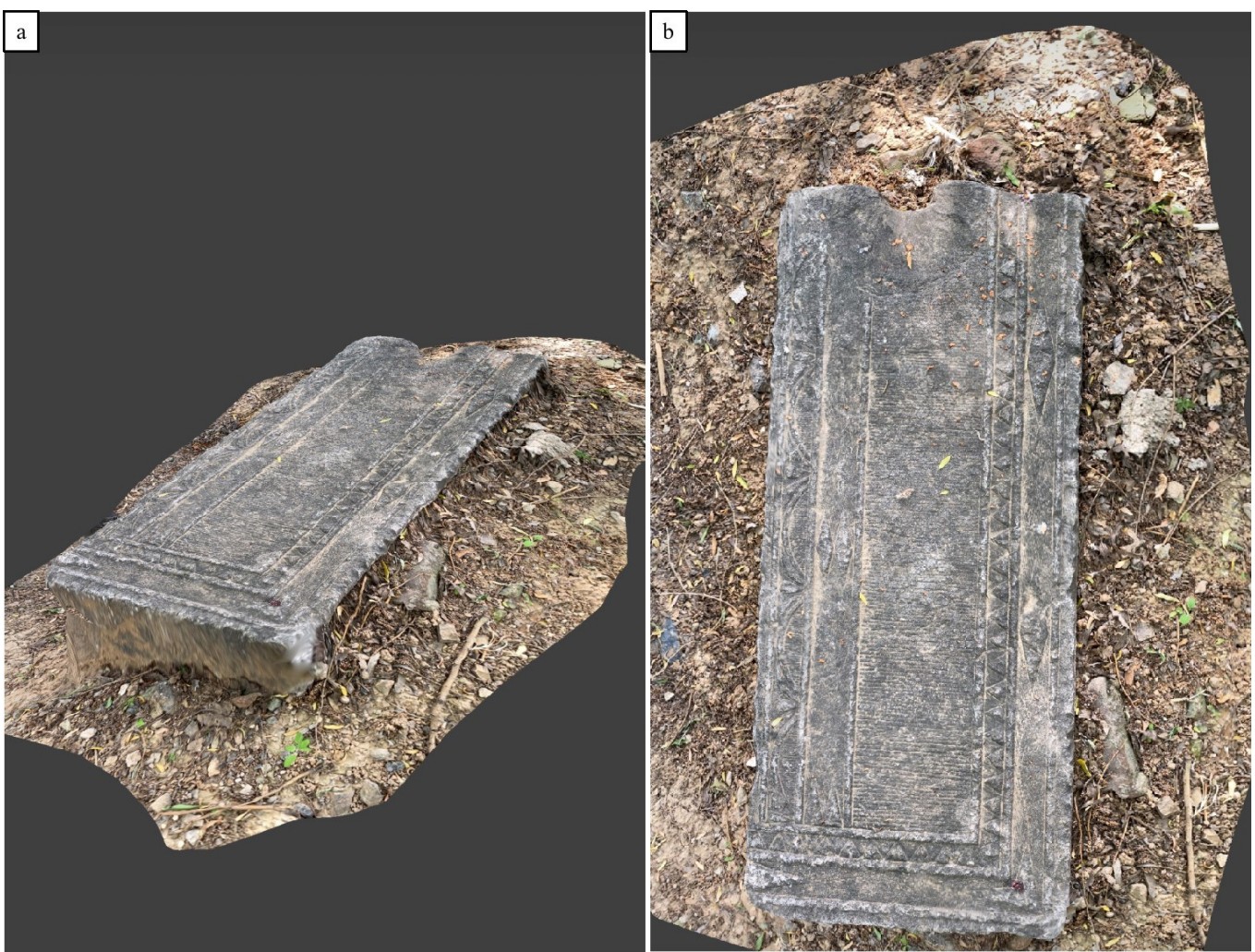

**Figure 6.** Three-dimensional modeling of Han Dynasty stone relief stored outdoors. Note the surface weathering under natural conditions. (**a**,**b**) Presentation from different perspectives.

Prior to the archaeological excavation, tomb no. 2 had already lost its burial mound, and it suffered severe damage. The 3D model can record the structure of the tomb chambers. Tomb no. 2 consists of a tomb door, a front chamber, a middle chamber, a back chamber, and a cloister, in which the tops of the front and middle chambers have no stone slabs, and the walls of the chamber have a layer of lime mud and the floor has a flat layer of stone slabs. Such scene materials can play many roles in historical and cultural research and revitalization. Firstly, it clearly preserves the shape and size of the chamber and digitally records the structural information of the chamber, which helps its research to be widely carried out and utilized. Secondly, such a scene record helps to carry out internet science popularization and propaganda, and can enrich the way of exhibiting cultural relics and monuments. Additionally, such a digital scene can be used as scene materials for games or virtual reality tourism.

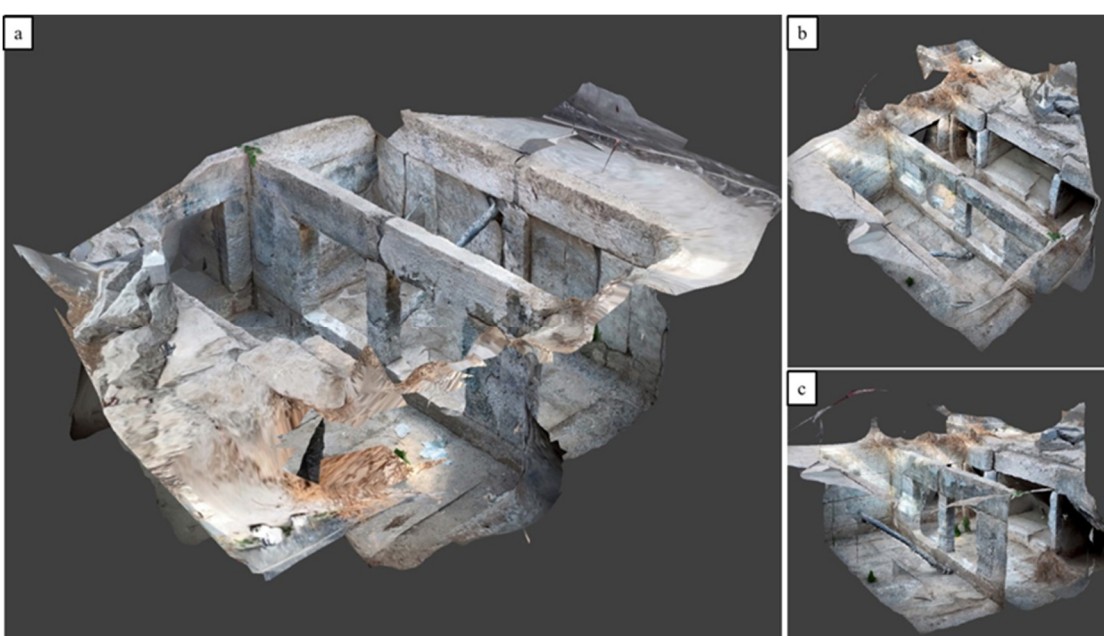

**Figure 7.** Three-dimensional modeling of the chamber structure of tomb no. 2. (**a–c**) Presentation of 3D model from different perspectives.

*4.4. Other Three-Dimensional Sculptures*

Ancient Chinese funerary relics often have other three-dimensional sculptures, such as stone monuments, stone queues, animal carvings, and figure carvings. Previous studies have focused on the flat-carved Han relief sculptures, overlooking the importance of the three-dimensional sculptural stone beasts found in divine paths. However, in the layout of Han Dynasty mausoleums, Shinto stone animals were an indispensable part. The Shinto stone animals are also called "stone elephants of life", which symbolize eternal life with eternal images of stone carvings. They appear in pairs on both sides of the Shinto path, escorting souls through life and death. It is the boundary marker of the geographic space of the family cemetery and the transition between the yin and yang of the living and the dead in the Han Dynasty funeral rituals, playing the role of guardianship and protection for the whole mausoleum. The research team selected the stone lion sculpture from the Wuliang Ancestral Hall for 3D modeling (Figure 8). This stone lion is the first ancient Chinese sculpture in which the lion figure appears. Lions were introduced to China through Central Asia during the Han Dynasty, bearing witness to cultural exchanges from two thousand years ago and possessing significant historical, cultural, and artistic value. The carving style differs greatly from the lion carvings of later dynasties. The stone lion is a three-dimensional round carving, with its head raised upward, tongue curled, and limbs stepping forward, which is quite dynamic and beautiful. The overall style of the stone lion is majestic, but the details are delicate and vivid, especially the head and neck with line carving techniques to show the curly mane and the base is decorated with beautiful curves and rhombus patterns around, the whole stone carving is both simple and natural, and exquisite and elegant.

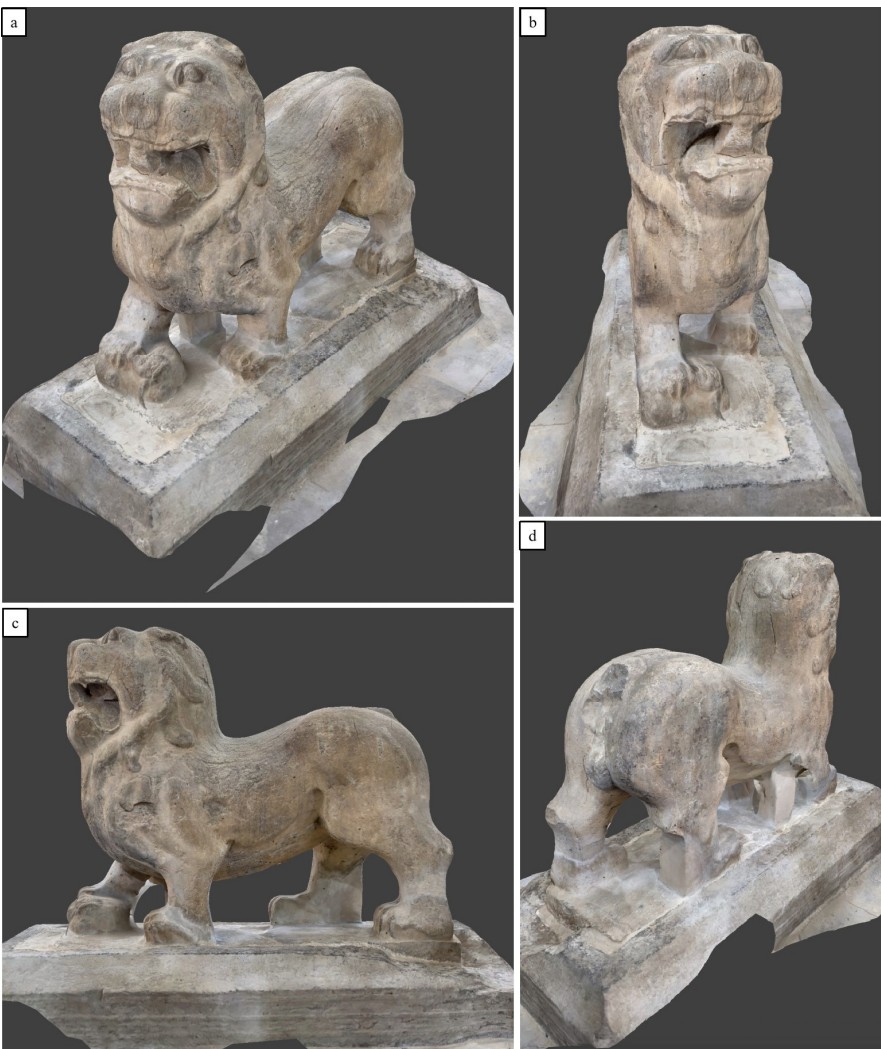

**Figure 8.** Three-dimensional modeling of stone lion sculpture. (**a–d**) Presentation of 3D model from different perspectives.

### 4.5. Significance of 3D Digital Modeling in the Conservation

Although scholars have paid attention to the collection and conservation of Han Dynasty stone reliefs, the preservation of related information still relies on the cultural relics themselves for a long time. In the four stages of conservation and research on Han Dynasty stone reliefs in China, rubbing technology has long played an important role in information preservation and dissemination. However, our research shows that traditional rubbing methods can cause damage to cultural relics, accelerate surface weathering, and ultimately cause information loss. In 2011, the National Administration of Cultural Heritage of China (NCHA) issued management measures for the reproduction and rubbing of cultural relics, limiting the rubbing of cultural relics. In 2022, the NCHA further issued a notice restricting the rubbing of stone cultural relics and prohibiting the sale of rubbings in order to protect cultural relics. Thus, a practical problem has emerged that conventional information extraction methods are no longer applicable, therefore, there is an urgent need to develop new methods and workflow for information extraction and conservation. From the research team's investigation, the new workflow needs to meet the following requirements: (1) it will not cause further damage to cultural relics; (2) it can effectively obtain three-dimensional structural information; (3) it can serve as a basis for further processing and utilization. The case study of the Wuliang Ancestral Hall revealed the role that 3D digital modeling can play in the conservation of Han Dynasty stone reliefs,

as well as the significance of digitizing cultural relics of different protection status and types and enriching future utilization approaches. The other issues in the conservation of stone cultural heritage are gradually receiving attention, which will provide more scientific guidance for the sustainable protection of this type of cultural heritage [21,22].

## 5. Work-Flow to Guide Sustainable Digital Conservation of Han Dynasty Stone Reliefs

More and more digital or artificial intelligence technologies are being applied in research related to cultural heritage protection [21–27]. The research team has carried out filed research, expert interviews, 3D scanning and modeling, and 3D printing. Based on the aforementioned research and experiments in 3D scanning, modeling, and replica production, the research team has analyzed the significance of 3D digital modeling for the conservation and potential revitalization of different types of Han pictograph stone-related cultural relics and sites. As Han Dynasty stone reliefs are susceptible to natural weathering or human destruction due to their shallow material and carvings, digital preservation can permanently preserve their digital versions for cultural heritage research and adaptive use [28]. At the same time, the combination of technologies such as 3D printing and 3D laser engraving allows for the physical reproduction of Han Dynasty stone reliefs and other cultural relics. Three-dimensional digital modeling can quickly preserve and record information about the scenes of Han Dynasty stone reliefs and related cultural relics and sites.

Based on the aforementioned research and expert advice, we propose the following workflow to guide the sustainable digital conservation of Han Dynasty stone reliefs (Figure 9). The first step is to obtain three-dimensional point cloud data of Han Dynasty stone reliefs or other cultural relics and sites through three-dimensional scanning. Where necessary, this can be further combined with rock mineralogy testing techniques such as X-ray Diffraction (XRD) to fuse other information such as their material with the 3D data. For surface images, texts, and ornaments of significant value, further 2D image processing can be carried out to extract image elements and data. For special types of sites such as burial chambers, scene information can be obtained through 3D scanning. Based on the scanned information a 3D digital model of the artefact itself as well as the scene is created, forming a dataset of that artefact, which is backed up and saved. Visualization models are produced based on the digitized models and used to enrich the presentation avenues of the artifacts. Based on the dataset, multiple avenues of revitalization can be carried out on Han Dynasty stone reliefs. Two-dimensional images and other information can be combined with AI models to carry out training, which in turn provides the basis for the production of AI-based works with the artistic style of Han Dynasty stone reliefs. Digitized models are easier to disseminate and more suitable for application in conjunction with internet technology development. The models can be used in the fields of games, animation, cultural and artistic works, etc. On the basis of the data and visualization models, further applications can be developed by combining 3D imaging, 3D printing, virtual reality, and other technologies. For example, 3D printing can be used to replicate physical models for replacing the originals for topography or for scientific research and display in order to avoid damage to the original artifacts. Based on such a workflow, an optimized conservation path can be established from digital preservation to sustainable revitalization. Compared to the current conservation status quo, such a conservation path greatly reduces the cost of cultural use and dissemination and increases the efficiency of heritage information conservation and use. Therefore, we suggest that whether it is a Han painting stone or other related cultural relics or monuments, they should first be digitally preserved in 3D when they are unearthed, and on this basis, information about the relics can be extracted, in-depth research can be carried out, and 3D models can be used for better science research and display.

Through the investigation, analysis and case study, a workflow has been formed to further guide the digital protection and revitalization of Han Dynasty stone reliefs. This

new set of technological processes integrates non-destructive recording, three-dimensional structural characterization, and digital modeling that can be further edited and utilized. The new workflow combines multiple digital technologies, providing a more flexible solution for the conservation and utilization of Han Dynasty stone reliefs. It meets the three requirements for the new workflow under the current conservation status and technological background. Compared with the current studies of new digital technology that are applied in the field of cultural resource conservation [23–31], this study focuses more on providing framework support for current obstacles. Compared to a specific technology application, the lack of a guiding workflow and framework for sustainable conservation is the core issue that restricts digital recording and utilization. Note that this is also a key factor that restricts the further utilization and development (revitalization) of the value of cultural heritage. What is urgently needed for the sustainable conservation and revitalization of Han Dynasty stone reliefs is the use of digital technologies and their combination that provide the foundation for protection, recording, and utilization [32]. These technologies and their combinations can reasonably constitute the workflow for cultural resource management with different protection status and characteristics, and then scientifically provide guidance for specific protection and utilization work.

In this study, the current status of the protection of Han Dynasty stone reliefs in Xuzhou area was investigated, which provides a basis for identifying the problems in current conservation and revitalization. Although scholars have been paying attention to Han Dynasty stone reliefs for a long time, there are still significant problems in its conservation and revitalization [32]. Since the information of image and calligraphy was recorded through three-dimensional surface carvings, natural weathering or human factors can cause damage of structure and loss of information. The speed of information loss is astonishing and irreversible. From the Song Dynasty to the present, conservation and research on Han Dynasty stone reliefs in China can be divided into four stages, and is currently transitioning from a comprehensive research stage to a digital protection stage. The investigate results show that, conventional methods of extracting information from Han Dynasty stone reliefs are limited to two-dimensional information and can cause damage to surface structure. The lack of digital technology and workflows or standards during conservation further restricts the utilization and revitalization of these cultural heritage. Determining how to utilize these aspects of cultural heritage to generate new value is a current research hotspot [33–35]. This puts forward new requirements for the protection of Han Dynasty stone reliefs, that is, to not only protect cultural heritage and its recorded information, but also further utilize its information. The case study of the Wuliang Ancestral Hall demonstrates how digital technology can play a role in the conservation and revitalization of Han Dynasty stone reliefs and related cultural relics or scenes in different preservation status. To provide solutions to the existing gaps, a work-flow including 3D scanning, data collection and processing, 3D modeling, visualization and information utilization is proposed. This workflow can provide scientific guidance for the sustainable conservation and revitalization of Han Dynasty stone reliefs in the research area to a certain extent.

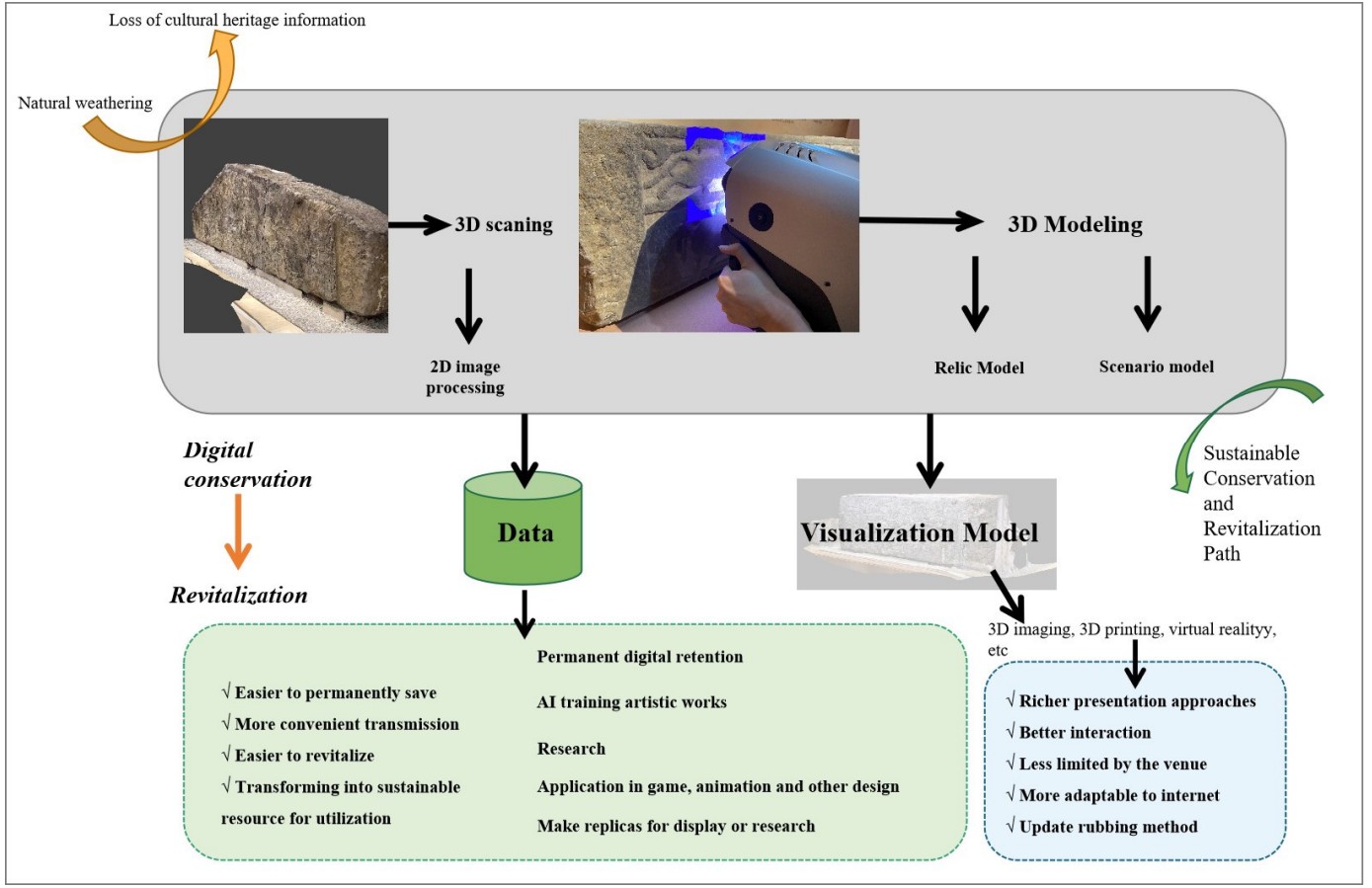

**Figure 9.** Work-flow to guide sustainable digital conservation and revitalization of Han Dynasty stone reliefs.

## 6. Conclusions

(1) The conservation and research on Han Dynasty stone reliefs in China can be divided into four stages. Currently, the conservation and utilization for the Han Dynasty stone reliefs in the study area are mainly through conservation institutions such as museums for storage. Influenced by the material and depth of carving, many Han portrait stones suffer from information loss, but there is still relatively little digital conservation and utilization of Han portrait stones and related cultural relics and sites.

(2) Conventional information extraction methods can cause damage to cultural relics and cannot meet the new needs of digital cultural utilization. The significance and function of 3D digital modeling for the conservation and revitalization of cultural relics with different type and preservation status was analyzed, indicating that utilizing a high-precision model and its data can be a good record of the information contained in the cultural relics and provide more potential avenues for their revitalization.

(3) Conventional protection and information extraction methods cannot meet the new requirements of sustainable conservation and revitalization. Therefore, three new requirements were proposed for future standards or workflows of sustainable conservation and revitalization including no further damage to cultural relics, acquisition and preservation capabilities for three-dimensional structural information, and potential for further processing and utilization. The case study of the Wuliang Ancestral Hall demonstrates the potential role and significance of 3D digital modelling in the conservation and revitalization of Han Dynasty stone reliefs.

(4) This study proposed a new workflow to guide sustainable digital conservation of Han Dynasty stone reliefs, which includes the steps of 3D scanning, data collection

and processing, 3D modeling, visualization and information utilization. Based on this workflow, an optimal conservation pathway from digital preservation to sustainable revitalization can be established. This study proposed existing problems and potential solutions for the conservation and revitalization of Han Dynasty stone reliefs, which can provide beneficial assistance and guidance for promoting the quality of cultural heritage protection in this field.

**Author Contributions:** Conceptualization, D.Z. and D.L.; methodology, D.Z., C.L. and J.H.; software, D.Z., J.H. and H.C.; validation, D.Z., X.Z. (Xinyue Zhang) and C.L.; formal analysis, X.Z. (Xinyue Zhang) and Y.D.; investigation, D.Z. and C.L.; resources, D.Z.; data curation, D.L. and X.Z. (Xiaoyue Zhai); writing—original draft preparation, D.Z. and X.Z. (Xiaoyue Zhai); writing—review and editing, D.Z., X.Z.(Xinyue Zhang) and J.H.; visualization, D.Z. and H.C.; supervision, D.Z. and P.L.; project administration, D.Z. and X.Z. (Xiaoyue Zhai); funding acquisition, D.Z. and C.L. All authors have read and agreed to the published version of the manuscript.

**Funding:** This work was supported by the Xuzhou Hantang Public Welfare Development Center funding project (No. 2021360004), National College Students' innovation and entrepreneurship training program (No. 202210290429E), Provincial College Student Innovation Training Project of Jiangsu Province (No. 202210290418H).

**Institutional Review Board Statement:** Not applicable.

**Informed Consent Statement:** Not applicable.

**Data Availability Statement:** The data presented in this study are available on request from the authors.

**Acknowledgments:** The authors acknowledge the editors and reviewers for their help. The authors extend their gratitude to Jiaming Zhang and Xin Guan from China University of Mining and Technology (CUMT) for their work in field research. The author Difei Zhao is particularly grateful for the support from the "Qihang" project of CUMT and the "Energy and Environment Youth Talent Training Program" of the China Environmental Protection Foundation, the China Energy Society, and the Beijing Energy and Environment Society.

**Conflicts of Interest:** The authors declare no conflict of interest.

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
