# Peer review of "3D Digital Modeling as a Sustainable Conservation and Revitalization Path for the Cultural Heritage of Han Dynasty Stone Reliefs"

_sustainability, doi:10.3390/su151612487_

Round 1

Reviewer 1 Report

The paper concentrates on important aspects with regard to the preservation , conservation, and "revitalization"  of the Han-Dynasty stone reliefs, in a case study on the material from the Xuzhou area that can be seen as a particular  part of  Chinese Han-Dynasty sculpture art. The authors deal with  the irreversible damages on them caused by weathering factors and human intervention (e.g., rubbing). To work against these problems, an application of three-dimensional digital technology is proposed that will guarantee information preservation.  The given situation and the possibilities of problem solving are well described, in connection with a review on the development of research on the stone reliefs. Based on tests with 3D scanning, modeling, and printing  of the stone reliefs, the authors are able to propose a workflow to proceed with a sustainable digital conservation in a successful way. Summarizing, one can emphasize that the article deals, on the one hand, with a particularly relevant part of Chinese tangible cultural heritage. On the other hand,  it offers interesting ways and well thought over proposals to solve the problems of conservation, preservation, research, and display with the help of 3D models. 

Author Response

Dear Editors and Reviewers:

Thank you for your letter and for the reviewers’ comments regarding the manuscript titled 3D Digital Modeling as a Sustainable Conservation and Revitalization Path for the Cultural Heritage of Han Dynasty Stone Reliefs. Thank you very much for your time spent in reading our paper, providing critical and meaningful comments. We have taken every effort to modify the manuscript in light of reviewers’ comments and queries. We highly appreciate the hard work and cooperation of the Editors/Reviewers’, and hope that the modified manuscript will be up to the standards of Sustainability.

We invite you to read detailed responses to the following comments, the modified content has been highlighted in green font.

To reviewer1:

The paper concentrates on important aspects with regard to the preservation , conservation, and "revitalization"  of the Han-Dynasty stone reliefs, in a case study on the material from the Xuzhou area that can be seen as a particular  part of  Chinese Han-Dynasty sculpture art. The authors deal with  the irreversible damages on them caused by weathering factors and human intervention (e.g., rubbing). To work against these problems, an application of three-dimensional digital technology is proposed that will guarantee information preservation.  The given situation and the possibilities of problem solving are well described, in connection with a review on the development of research on the stone reliefs. Based on tests with 3D scanning, modeling, and printing  of the stone reliefs, the authors are able to propose a workflow to proceed with a sustainable digital conservation in a successful way. Summarizing, one can emphasize that the article deals, on the one hand, with a particularly relevant part of Chinese tangible cultural heritage. On the other hand,  it offers interesting ways and well thought over proposals to solve the problems of conservation, preservation, research, and display with the help of 3D models. 

Response:

Thank you to the reviewers for their recognition of the research. We will continue to carry out subsequent research work under the guidance of your opinions.

Reviewer 2 Report

I think the paper is very nice. As the authors said: Faced with the challenge of preserving and protecting historical heritage and its information, digital technology can become an important alternative method, providing new approaches for their sustainable conservation and revitalization. I do hope more related papers can be published in the future.

nice

Author Response

Dear Editors and Reviewers:

Thank you for your letter and for the reviewers’ comments regarding the manuscript titled 3D Digital Modeling as a Sustainable Conservation and Revitalization Path for the Cultural Heritage of Han Dynasty Stone Reliefs. Thank you very much for your time spent in reading our paper, providing critical and meaningful comments. We have taken every effort to modify the manuscript in light of reviewers’ comments and queries. We highly appreciate the hard work and cooperation of the Editors/Reviewers’, and hope that the modified manuscript will be up to the standards of Sustainability.

We invite you to read detailed responses to the following comments, the modified content has been highlighted in green font.

To reviewer2:

I think the paper is very nice. As the authors said: Faced with the challenge of preserving and protecting historical heritage and its information, digital technology can become an important alternative method, providing new approaches for their sustainable conservation and revitalization. I do hope more related papers can be published in the future.

Response:

Thank you to the reviewers for their recognition of the research. We will continue to carry out subsequent research work under the guidance of your opinions.

Reviewer 3 Report

1.                  Abstract should reflect the background knowledge on the problem addressed and is need to be added.

2.                  In Introduction section, the drawbacks of each conventional technique should be described clearly.

3.                  Introduction section can be extended to add the issues with respect to existing work.

4.                  Properly define the motivation behind the work.

5.                  Authors can use latest related works from reputed journals like IEEE/ACM Transactions, Elsevier, Inderscience, Springer, Taylor & Francis etc. and write the references in proper format, from year 2022-2023

6.      Abstract and Conclusion should be concise yet. But should give complete overview of the work and study.

7.      The authors seem to disregard or neglect some important finding in results that have been achieved in paper. So, elaborate and explain the results in more details.

8.      Improve the results and discussion section in paragraph.

Minor editing of English language required

Author Response

Dear Editors and Reviewers:

Thank you for your letter and for the reviewers’ comments regarding the manuscript titled 3D Digital Modeling as a Sustainable Conservation and Revitalization Path for the Cultural Heritage of Han Dynasty Stone Reliefs. Thank you very much for your time spent in reading our paper, providing critical and meaningful comments. We have taken every effort to modify the manuscript in light of reviewers’ comments and queries. We highly appreciate the hard work and cooperation of the Editors/Reviewers’, and hope that the modified manuscript will be up to the standards of Sustainability.

We invite you to read detailed responses to the following comments, the modified content has been highlighted in green font.

To reviewer 3:

  1. Abstract should reflect the background knowledge on the problem addressed and is need to be added.

Response:

Many thanks for this advice. We have added the background knowledge on the problem addressed in the abstract as follows:

“Cultural relics and historical sites serve as carriers of cultural, historical, and artistic information. However, any damage incurred by these cultural relics can result in the loss of information, consequently impacting sustainable conservation and revitalization of the cultural heritage. ”

  1. In Introduction section, the drawbacks of each conventional technique should be described clearly.

Response:

Thanks for this thoughtful advice. We have added the description of the drawbacks of each conventional technique as follows:

“At present, the protection of Han dynasty stone reliefs primarily relies on centralized collection and protection by museums and archaeological research institutions. Only a small portion of the original sites receives protection. However, there appears to be a lack of attention toward addressing surface weathering, damage, and the continuous loss of information. In the protection of information on stone reliefs, rubbing and imaging are the main methods employed. However, the rubbing method can cause additional surface damage, and it is difficult to preserve the three-dimensional structural information of the surface of stone reliefs through imaging. These conventional protection methods have become increasingly difficult to adapt to higher protection requirements and fail to provide a digital information foundation for activating cultural heritage.”

  1. Introduction section can be extended to add the issues with respect to existing work.

Response:

Thanks for this thoughtful advice. We have added the issues related to existing work in the introduction section as follows:

“In recent years, the Xuzhou area has witnessed continuous discoveries of Han Dynasty stone reliefs, presenting both new opportunities for cultural and historical research and fresh challenges concerning effective sustainable conservation and revitalization. Although museums and other institutions have effectively protected cultural relics, issues such as surface damage and information loss persist. Conventional methods of cultural relic protection are no longer sufficient to support the revitalization of these cultural relics. In addition, the utilization of cultural heritage information to foster cultural innovation in the future demands further research. To address these concerns, a comprehensive investigation of the current situation is essential. This will lay the groundwork for establishing a more systematic work programme and workflow based on advanced technological methods. This approach aims to achieve three-dimensional digital conservation and revitalization of the cultural heritage of Han Dynasty stone reliefs. Through the preliminary investigate on the conservation and revitalization of Han Dynasty stone reliefs in Xuzhou area, the research team found that the lack of effective innovative technological paths for information preservation and utilization has significantly restricted the sustainable protection and intangible resource utilization of these cultural heritage. ”

  1. Properly define the motivation behind the work.

Response:

Thanks for this thoughtful advice. We have added the motivation behind the work in the introduction section as follows:

“Through the preliminary survey on the conservation and revitalization of Han Dynasty stone reliefs in Xuzhou area, the research team found that the lack of effective innovative technological paths for information preservation and utilization has significantly restricted the sustainable protection and intangible resource utilization of these cultural heritage. ”

  1. Authors can use latest related works from reputed journals like IEEE/ACM Transactions, Elsevier, Inderscience, Springer, Taylor & Francis etc. and write the references in proper format, from year 2022-2023

Response:

Thanks for this thoughtful advice. We have cited more latest related works. 

More and more digital or artificial intelligence technologies are being applied in research related to cultural heritage protection [21-27].

21.Pan, J.; Li, L.; Hiroshi,Y.; Kyoko, H.; Fadjar, I.T.; Brahmantara; Satoshi, T. 3D reconstruction of Borobudur reliefs from 2D monocular photographs based on soft-edge enhanced deep learning. ISPRS Journal of Photogrammetry and Remote Sensing, 2022, 183, 439-450.

22.Amirhosein, S.; Margarita, S.; Sevasti, T.; Andreas, G.; Vagelis, P.; Mahdi, K. 3D simulation models for developing digital twins of heritage structures: challenges and strategies. Procedia Structural Integrity, 2022, 37, 314-320.

23.Martina, M.; Arianna, L.; Panagiotis, I.; Lisa, P.; Antonella, T.; Alex, Z.; Elena, S.; Antonio, M. Environmental and economic sustainability in cultural heritage preventive conservation: LCA and LCC of innovative nanotechnology-based products. Cleaner Environmental Systems, 2023, 9, 100-124.

24.Wang, B.S.; Gamze, D.; Theo, A. A structural equation model to analyze theuse of a new multi media platform forincreasing awareness of cultural heritage.  Frontiers of Architectural Research, 2023, 12, 509-522.

25.Wang, X.; Zhu, C.S.; Hu, Y.L.; Zhang, Z.; Zhang, B.J. Development and application of cinnamaldehyde-loaded halloysite nanotubes for the conservation of stone cultural heritage. Applied clay science, 2023, 236.

26.Victoria, A.C. From 3D point clouds to HBIM: Application of Artificial Intelligence in Cultural Heritage. Automation in construction, 2023, 152.

  1. 2Jia,S.Z.; Liao, Y.; Xiao, Y.Q.; Zhang, B.; Meng, X.B.; Qin,K. Conservation and management of Chinese classical royal garden heritages based on 3D digitalization - A case study of Jianxin courtyard in Jingyi garden in fragrant hills. Journal of cultural heritage, 2022, 58, 102-111.

  1. Abstract and Conclusion should be concise yet. But should give complete overview of the work and study.

Response:

Thanks for this thoughtful advice. We have revised Abstract and Conclusion to be more concise and comprehensive:

“In this study, a systematic investigation was carried out to study the current situation and existing problems related to the protection of Han Dynasty stone reliefs. Additionally, a case study was conducted, using the Wuling Ancestral Hall (Wuliang Shrine) as an example, to explore the integration of 3D digital technology as a new sustainable approach. The results show that natural weathering and conventional techniques have caused irreversible information loss. Thus, adopting a three-dimensional digital perspective is crucial when considering the information preservation and revitalization of Han Dynasty stone reliefs. To achieve this, 3D digital models of representative stone reliefs, tomb chambers, and other sculptures from the Wuliang Ancestral Hall were established. These models provide new paths for accurately recording 3D information and better utilizing cultural heritage. Faced with the challenge of preserving historical heritage and its associated information, a workflow including 3D scanning, data collection and processing, 3D modeling, visualization, and information utilization is proposed. This approach offers new approaches for sustainable conservation and revitalization of Han Dynasty stone reliefs. ”

“6.1 The conservation and research on Han Dynasty stone reliefs in China can be divided into four stages. Currently, the conservation and utilization for the Han dynasty stone reliefs in the study area are mainly through conservation institutions such as museums for storage. Influenced by the material and depth of carving, many Han portrait stones suffer from information loss, but there is still relatively little digital conservation and utilization of Han portrait stones and related cultural relics and sites.

6.2 Conventional information extraction methods can cause damage to cultural relics  and cannot meet the new needs of digital cultural utilization. The significance and function of 3D digital modeling for the conservation and revitalization of cultural relics with different type and preservation status was analyzed, indicating that the high-precision model and its data can be a good record of the information contained in the cultural relics and provide more potential avenues for their Revitalization.

6.3 This study proposed a workflow to guide sustainable digital conservation of Han Dynasty stone reliefs, which includes the steps of 3D scaning, data collection and processing, 3D modeling, visualization and information utilization. Based on this workflow, an optimal conservation pathway from digital preservation to sustainable revitalization can be established.”

  1. The authors seem to disregard or neglect some important finding in results that have been achieved in paper. So, elaborate and explain the results in more details.

Response:

Thanks for this thoughtful advice. We have discussed about this problem and added more details on the the neglected findings in results.

“Although scholars have paid attention to the collection and conservation of Han Dynasty stone reliefs, the preservation of related information still relies on the cultural relics themselves for a long time. In the four stages of conservation and research on Han Dynasty stone reliefs in China, rubbing technology has long played an important role in information preservation and dissemination. But our research shows that traditional rubbing methods can cause damage to cultural relics, accelerate surface weathering, and ultimately cause information loss. In 2011, the National Administration of Cultural Heritage of China (NCHA) issued management measures for the reproduction and rubbing of cultural relics, limiting the rubbing of cultural relics. In 2022, NCHA further issued a notice restricting the rubbing of stone cultural relics and prohibiting the sale of rubbings in order to protect cultural relics. Thus, a practical problem has emerged that conventional information extraction methods are no longer applicable, therefore, there is an urgent need to develop new methods and workflow for information extraction and conservation. From the research team's investigation, the new workflow needs to meet the following requirements: 1) it will not cause further damage to cultural relics; 2) it can effectively obtain three-dimensional structural information; 3) it can serve as a basis for further processing and utilization. The case study of the Wuliang Ancestral Hall revealed the role that 3D Digital Modeling can play in the conservation of Han Dynasty stone reliefs, as well as the significance of digitizing cultural relics of different protection status and types and enriching future utilization approaches. ”

“In this study, the current status of the protection of Han Dynasty stone reliefs in Xuzhou area was investigated, which provides a basis for identifying the problems in current conservation and revitalization. Although scholars have been paying attention to Han Dynasty stone reliefs for a long time, there are still significant problems in its conservation and revitalization. Due to the information of image and calligraphy was recorded through three-dimensional surface carvings, natural weathering or human factors can cause irreversible damage of structure and loss of information. The speed of information loss is astonishing and irreversible. From the Song Dynasty to the present, conservation and research on Han Dynasty stone reliefs in China can be divided into four stages, and is currently transitioning from a comprehensive research stage to a digital protection stage. The investigate results show that, conventional methods of extracting information from Han dynasty stone reliefs are limited to two-dimensional information and can cause damage to surface structure. The lack of digital technology and workflows or standards during conservation further restricts the utilization and revitalization of these cultural heritage. How to utilize these cultural heritage to generate new value is a current research hotspot. This puts forward new requirements for the protection of Han Dynasty stone reliefs, that is, to not only protect cultural heritage and its recorded information, but also further utilize its information. The case study of the Wuliang Ancestral Hall demonstrates how digital technology can play a role in the conservation and revitalization of Han Dynasty stone reliefs and related cultural relics or scenes in different preservation states. To provide solutions to the existing gaps, a work-flow including 3D scaning, data collection and processing, 3D modeling, visualization and information utilization is proposed. This work flow can provide scientific guidance for the sustainable conservation and revitalization of Han Dynasty stone reliefs in the research area to a certain extent. ”

  1. Improve the results and discussion section in paragraph.

Response:

Thanks for this thoughtful advice. We have improve the results and discussion section in paragraph.

“In this study, the current status of the protection of Han Dynasty stone reliefs in Xuzhou area was investigated, which provides a basis for identifying the problems in current conservation and revitalization. Although scholars have been paying attention to Han Dynasty stone reliefs for a long time, there are still significant problems in its conservation and revitalization. Due to the information of image and calligraphy was recorded through three-dimensional surface carvings, natural weathering or human factors can cause irreversible damage of structure and loss of information. The speed of information loss is astonishing and irreversible. From the Song Dynasty to the present, conservation and research on Han Dynasty stone reliefs in China can be divided into four stages, and is currently transitioning from a comprehensive research stage to a digital protection stage. The investigate results show that, conventional methods of extracting information from Han dynasty stone reliefs are limited to two-dimensional information and can cause damage to surface structure. The lack of digital technology and workflows or standards during conservation further restricts the utilization and revitalization of these cultural heritage. How to utilize these cultural heritage to generate new value is a current research hotspot. This puts forward new requirements for the protection of Han Dynasty stone reliefs, that is, to not only protect cultural heritage and its recorded information, but also further utilize its information. The case study of the Wuliang Ancestral Hall demonstrates how digital technology can play a role in the conservation and revitalization of Han Dynasty stone reliefs and related cultural relics or scenes in different preservation states. To provide solutions to the existing gaps, a work-flow including 3D scaning, data collection and processing, 3D modeling, visualization and information utilization is proposed. This work flow can provide scientific guidance for the sustainable conservation and revitalization of Han Dynasty stone reliefs in the research area to a certain extent. ”

Reviewer 4 Report

3D Digital Modelling as a Sustainable Conservation and Revitalization Path for the Cultural Heritage of Han Dynasty Stone 3 Reliefs

·        A Research article is expected to have a novel methodology,

·        A real problem to be solved.

·        Discuss the Research Gap in detail

·        Find solutions to these gaps

·        Make a novel Methodology

·        Discuss the performance Evaluation metrics Based on these metrics the results are judged.

·        A research paper is expected to have a good result with justifications

·        Finally, the results obtained should be compared with the state-of-the-art techniques.

·        Based on all the above points I could not find these in the current paper. 

No Comments

Author Response

Dear Editors and Reviewers:

Thank you for your letter and for the reviewers’ comments regarding the manuscript titled 3D Digital Modeling as a Sustainable Conservation and Revitalization Path for the Cultural Heritage of Han Dynasty Stone Reliefs. Thank you very much for your time spent in reading our paper, providing critical and meaningful comments. We have taken every effort to modify the manuscript in light of reviewers’ comments and queries. We highly appreciate the hard work and cooperation of the Editors/Reviewers’, and hope that the modified manuscript will be up to the standards of Sustainability.

We invite you to read detailed responses to the following comments, the modified content has been highlighted in green font.

To reviewer4:

A Research article is expected to have A real problem to be solved.

Response:

Many thanks for this advice. We have added the problem and background knowledge in the Introduction section as follows:

“Cultural relics and historical sites serve as carriers of cultural, historical, and artistic information. However, any damage incurred by these cultural relics can result in the loss of information, consequently impacting sustainable conservation and revitalization of the cultural heritage. ”

“In recent years, the Xuzhou area has witnessed continuous discoveries of Han Dynasty stone reliefs, presenting both new opportunities for cultural and historical research and fresh challenges concerning effective sustainable conservation and revitalization. Although museums and other institutions have effectively protected cultural relics, issues such as surface damage and information loss persist. Conventional methods of cultural relic protection are no longer sufficient to support the revitalization of these cultural relics. In addition, the utilization of cultural heritage information to foster cultural innovation in the future demands further research. To address these concerns, a comprehensive investigation of the current situation is essential. This will lay the groundwork for establishing a more systematic work programme and workflow based on advanced technological methods. This approach aims to achieve three-dimensional digital conservation and revitalization of the cultural heritage of Han Dynasty stone reliefs. Through the preliminary investigate on the conservation and revitalization of Han Dynasty stone reliefs in Xuzhou area, the research team found that the lack of effective innovative technological paths for information preservation and utilization has significantly restricted the sustainable protection and intangible resource utilization of these cultural heritage. ”

“The logic of methodology is to identify existing problems and research gaps, and design a new work-flow through case study to private potential new solutions for sustainable conservation and revitalization of Han Dynasty stone reliefs.”

  • Discuss the Research Gap in detailand Find solutions to these gaps

Response:

Many thanks for this advice. We have added more information and discussion on the research gaps and solutions in detail.

“Cultural relics and historical sites serve as carriers of cultural, historical, and artistic information. However, any damage incurred by these cultural relics can result in the loss of information, consequently impacting sustainable conservation and revitalization of the cultural heritage. ”

“At present, the protection of Han dynasty stone reliefs primarily relies on centralized collection and protection by museums and archaeological research institutions. Only a small portion of the original sites receives protection. However, there appears to be a lack of attention toward addressing surface weathering, damage, and the continuous loss of information. In the protection of information on stone reliefs, rubbing and imaging are the main methods employed. However, the rubbing method can cause additional surface damage, and it is difficult to preserve the three-dimensional structural information of the surface of stone reliefs through imaging. These conventional protection methods have become increasingly difficult to adapt to higher protection requirements and fail to provide a digital information foundation for activating cultural heritage.”

“In recent years, Han Dynasty stone reliefs have been continuously unearthed in the Xuzhou area. This brings new opportunities for cultural and historical research, but also raises new problems on how to effectively carry out sustainable conservation and revitalization. Although museums and other institutions have effectively protected cultural relics, problems of surface damage and information loss still exist, and conventional forms of cultural relic protection are no longer able to provide basic support for the revitalization of cultural relics. In addition, how to utilize cultural heritage information to promote cultural innovation in the future is also an important aspect worth further research. Therefore, on the basis of thorough investigation of the current situation, it is necessary to form a more systematic work programme and work-flow based on advanced technological methods in order to better achieve three-dimensional digital conservation and revitalization of the cultural heritage of Han Dynasty stone reliefs. Through the preliminary investigate on the conservation and revitalization of Han Dynasty stone reliefs in Xuzhou area, the research team found that the lack of effective innovative technological paths for information preservation and utilization has significantly restricted the sustainable protection and intangible resource utilization of these cultural heritage. ”

“Compared to a specific technology application, the lack of a guiding workflow and framework for sustainable conservation is the core issue that restricts digital recording and utilization. ”

“Although scholars have paid attention to the collection and conservation of Han Dynasty stone reliefs, the preservation of related information still relies on the cultural relics themselves for a long time. In the four stages of conservation and research on Han Dynasty stone reliefs in China, rubbing technology has long played an important role in information preservation and dissemination. But our research shows that traditional rubbing methods can cause damage to cultural relics, accelerate surface weathering, and ultimately cause information loss. In 2011, the National Administration of Cultural Heritage of China (NCHA) issued management measures for the reproduction and rubbing of cultural relics, limiting the rubbing of cultural relics. In 2022, NCHA further issued a notice restricting the rubbing of stone cultural relics and prohibiting the sale of rubbings in order to protect cultural relics. Thus, a practical problem has emerged that conventional information extraction methods are no longer applicable, therefore, there is an urgent need to develop new methods for information extraction and conservation. From the research team's investigation, the new technical method needs to meet the following requirements: 1) it will not cause further damage to cultural relics; 2) it can effectively obtain three-dimensional structural information; 3) it can serve as a basis for further processing and utilization. The case study of the Wuliang Ancestral Hall revealed the role that 3D Digital Modeling can play in the conservation of Han Dynasty stone reliefs, as well as the significance of digitizing cultural relics of different protection status and types and enriching future utilization approaches. ”

  • Make a novel Methodology

Response:

Many thanks for this advice. We have added more description of the methodology.

“Firstly, the team conducted systematic fieldwork and surveys work in the research area to gain a comprehensive understanding of the current situation and technical methods used in protecting Han Dynasty stone reliefs. During this process, the research team engaged in extensive communication with professional technical personnel from museums and related institutions, obtaining data and materials. On this basis, the research team selected the Wuliang Ancestral Hall (Wuliang Shrine) in Jining, Shandong Province as an example to conduct a case study. In the case study, the research team not only conducted research on the conservation status of Han stone reliefs and other cultural heritage in the Wuliang Ancestral Hall, but also attempted digital conservation of cultural heritage with different types and preservation status through technical methods such as 3D scanning, 3D modeling, and 3D printing. Finally, the research team combined the results of field research, expert interviews, 3D scanning, 3D modeling, and 3D printing to form a new workflow to better guide sustainable digital conservation and revitalization of Han Dynasty stone reliefs. The logic of the methodology is to identify existing problems and research gaps, then designing a new workflow through a case study to private potential new solutions for sustainable conservation and revitalization of Han Dynasty stone reliefs.”

  • Discuss the performance Evaluation metrics Based on these metrics the results are judged.

Response:

Many thanks for this advice. We have added the performance evaluation metrics and related discussion.

“From the research team's investigation, the new workflow needs to meet the following requirements: 1) it will not cause further damage to cultural relics; 2) it can effectively obtain three-dimensional structural information; 3) it can serve as a basis for further processing and utilization. The case study of the Wuliang Ancestral Hall revealed the role that 3D Digital Modeling can play in the conservation of Han Dynasty stone reliefs, as well as the significance of digitizing cultural relics of different protection status and types and enriching future utilization approaches. ”

“Through the investigation, analysis and case study, a workflow has been formed to further guide the digital protection and revitalization of Han Dynasty stone reliefs. This new set of technological processes integrates non-destructive recording, three-dimensional structural characterization, and digital modeling that can be further edited and utilized. The new workflow combines multiple digital technologies, providing a more flexible solution for the conservation and utilization of Han Dynasty stone reliefs. It meets the three requirements for the new workflow under the current conservation status and technological background. ”

  • A research paper is expected to have a good result with justifications
  • Finally, the results obtained should be compared with the state-of-the-art techniques.

 Response:

Many thanks for this advice. We have added more details of result with justifications and comparison with the state-of-the-art techniques.

“Although scholars have paid attention to the collection and conservation of Han Dynasty stone reliefs, the preservation of related information still relies on the cultural relics themselves for a long time. In the four stages of conservation and research on Han Dynasty stone reliefs in China, rubbing technology has long played an important role in information preservation and dissemination. But our research shows that traditional rubbing methods can cause damage to cultural relics, accelerate surface weathering, and ultimately cause information loss. In 2011, the National Administration of Cultural Heritage of China (NCHA) issued management measures for the reproduction and rubbing of cultural relics, limiting the rubbing of cultural relics. In 2022, NCHA further issued a notice restricting the rubbing of stone cultural relics and prohibiting the sale of rubbings in order to protect cultural relics. Thus, a practical problem has emerged that conventional information extraction methods are no longer applicable, therefore, there is an urgent need to develop new methods and workflow for information extraction and conservation. From the research team's investigation, the new workflow needs to meet the following requirements: 1) it will not cause further damage to cultural relics; 2) it can effectively obtain three-dimensional structural information; 3) it can serve as a basis for further processing and utilization. The case study of the Wuliang Ancestral Hall revealed the role that 3D Digital Modeling can play in the conservation of Han Dynasty stone reliefs, as well as the significance of digitizing cultural relics of different protection status and types and enriching future utilization approaches.”

“Compared with the current studies of new digital technology that applied in the field of cultural resource conservation [21-27], this study focuses more on providing framework support for current obstacles. Compared to a specific technology application, the lack of a guiding workflow and framework for sustainable conservation is the core issue that restricts digital recording and utilization. Note that this is also a key factor that restricts the further utilization and development (revitalization) of the value of cultural heritage. What is urgently needed for the sustainable conservation and revitalization of Han Dynasty stone reliefs is the digital technologies and their combination that provide the foundation for protection, recording, and utilization. These technologies and their combinations can reasonably constitute the workflow for cultural resource management with different protection status and characteristics, and then scientifically provide guidance for specific protection and utilization work. ”

“In this study, the current status of the protection of Han Dynasty stone reliefs in Xuzhou area was investigated, which provides a basis for identifying the problems in current conservation and revitalization. Although scholars have been paying attention to Han Dynasty stone reliefs for a long time, there are still significant problems in its conservation and revitalization. Due to the information of image and calligraphy was recorded through three-dimensional surface carvings, natural weathering or human factors can cause irreversible damage of structure and loss of information. The speed of information loss is astonishing and irreversible. From the Song Dynasty to the present, conservation and research on Han Dynasty stone reliefs in China can be divided into four stages, and is currently transitioning from a comprehensive research stage to a digital protection stage. The investigate results show that, conventional methods of extracting information from Han dynasty stone reliefs are limited to two-dimensional information and can cause damage to surface structure. The lack of digital technology and workflows or standards during conservation further restricts the utilization and revitalization of these cultural heritage. How to utilize these cultural heritage to generate new value is a current research hotspot. This puts forward new requirements for the protection of Han Dynasty stone reliefs, that is, to not only protect cultural heritage and its recorded information, but also further utilize its information. The case study of the Wuliang Ancestral Hall demonstrates how digital technology can play a role in the conservation and revitalization of Han Dynasty stone reliefs and related cultural relics or scenes in different preservation states. To provide solutions to the existing gaps, a work-flow including 3D scaning, data collection and processing, 3D modeling, visualization and information utilization is proposed. This work flow can provide scientific guidance for the sustainable conservation and revitalization of Han Dynasty stone reliefs in the research area to a certain extent. ”

Round 2

Reviewer 4 Report

I think the authors have revised the paper based on the comment. So, I suggest accepting the paper.

None

Author Response

Dear Editors and Reviewers:

Thank you for your letter and for the  comments regarding the manuscript titled 3D Digital Modeling as a Sustainable Conservation and Revitalization Path for the Cultural Heritage of Han Dynasty Stone Reliefs. Thank you very much for your time spent in reading our paper, providing critical and meaningful comments. We have taken every effort to modify the manuscript in light of reviewers’ comments and queries. We highly appreciate the hard work and cooperation of the Editors/Reviewers’, and hope that the modified manuscript will be up to the standards of Sustainability.

We invite you to read detailed responses to the following comments, the modified content has been highlighted in blue font.

To editor:

Revise the conclusion to highlight innovative findings and contributions to knowledge in the field.

Response:

Many thanks for this advice. We have added more information of innovative findings and contributions to knowledge in the field as follows:

“6.3 Conventional protection and information extraction methods cannot meet the new requirements of sustainable conservation and revitalization. Therefore, three new requirements were proposed for future standards or workflows of sustainable conservation and revitalization including no further damage to cultural relics, acquisition and preservation capabilities for three-dimensional structural information, and potential for further processing and utilization. The case study of the Wuliang Ancestral Hall demonstrates the potential role and significance of 3D digital modelling in the conservation and revitalization of Han Dynasty stone reliefs.  

6.4 This study proposed a new workflow to guide sustainable digital conservation of Han Dynasty stone reliefs, which includes the steps of 3D scaning, data collection and processing, 3D modeling, visualization and information utilization. Based on this workflow, an optimal conservation pathway from digital preservation to sustainable revitalization can be established. This study proposed existing problems and potential solutions for the conservation and revitalization of Han Dynasty stone reliefs, which can provide beneficial assistance and guidance for promoting the quality of cultural heritage protection in this field. ”

Introduce clear aspects of innovation and originality in the research methodology.

Response:

Many thanks for this advice. We have added more information of innovation and originality in the research methodology as follows:

“In methodology, this study innovatively designed a research process of status investigation, problem identification, case study, and experience summary. Meanwhile, due to the unique research objective, this study adopts a interdisciplinary approach to conduct research. ”

Introduce more related literature in photogrammetry and digital display. Currently the literature is limited

Response:

Many thanks for this advice. We have added more related literature in photogrammetry and digital display as follows:

The other issues in the conservation of stone cultural heritage are gradually receiving attention, which will provide more scientific guidance for the sustainable protection of this type of cultural heritage [21-22].

21.Huang, J.Z.; Zheng, Y.; Li, H. Study of internal moisture condensation for the conservation of stone cultural heritage. Journal of Cultural Heritage. 2022, 56, 1-9.

22.Castro, N.F.; Becerra, J.E.; Bellopede, R.; Marini, P.; Dino, G.A. Introduction to ‘natural stones and cultural heritage promotion and preservation’. Resources Policy. 2022, 78, 102775.

Compared with the current studies of new digital technology that applied in the field of cultural resource conservation [23-31]:

23.Pan, J.; Li, L.; Hiroshi,Y.; Kyoko, H.; Fadjar, I.T.; Brahmantara; Satoshi, T. 3D reconstruction of Borobudur reliefs from 2D monocular photographs based on soft-edge enhanced deep learning. ISPRS Journal of Photogrammetry and Remote Sensing, 2022, 183, 439-450.

24.Amirhosein, S.; Margarita, S.; Sevasti, T.; Andreas, G.; Vagelis, P.; Mahdi, K. 3D simulation models for developing digital twins of heritage structures: challenges and strategies. Procedia Structural Integrity, 2022, 37, 314-320.

25.Martina, M.; Arianna, L.; Panagiotis, I.; Lisa, P.; Antonella, T.; Alex, Z.; Elena, S.; Antonio, M. Environmental and economic sustainability in cultural heritage preventive conservation: LCA and LCC of innovative nanotechnology-based products. Cleaner Environmental Systems, 2023, 9, 100-124.

26.Wang, B.S.; Gamze, D.; Theo, A. A structural equation model to analyze theuse of a new multi media platform forincreasing awareness of cultural heritage.  Frontiers of Architectural Research, 2023, 12, 509-522.

27.Wang, X.; Zhu, C.S.; Hu, Y.L.; Zhang, Z.; Zhang, B.J. Development and application of cinnamaldehyde-loaded halloysite nanotubes for the conservation of stone cultural heritage. Applied clay science, 2023, 236.

28.Victoria, A.C. From 3D point clouds to HBIM: Application of Artificial Intelligence in Cultural Heritage. Automation in construction, 2023, 152.

29.Jia, S.Z.; Liao, Y.; Xiao, Y.Q.; Zhang, B.; Meng, X.B.; Qin,K. Conservation and management of Chinese classical royal garden heritages based on 3D digitalization - A case study of Jianxin courtyard in Jingyi garden in fragrant hills. Journal of cultural heritage, 2022, 58, 102-111.

30.Kutlu, I.; Soyluk, A. A comparative approach to using photogrammetry in the structural analysis of historical buildings. Ain Shams Engineering Journal. 2023, In Press.

31.María, HiguerasAna.; Isabel, CaleroFrancisco.; José, Collado-Montero. Digital 3D modeling using photogrammetry and 3D printing applied to the restoration of a Hispano-Roman architectural ornament. Digital Applications in Archaeology and Cultural Heritage. 2021, 20, e00179.

Although scholars have been paying attention to Han Dynasty stone reliefs for a long time, there are still significant problems in its conservation and revitalization [32].

32.Aleš, J.; Miran, E.; Igor, M.; Žiga, S.; Franc, S. Volumetric models from 3D point clouds: the case study of sarcophagi cargo from a 2nd/3rd century AD Roman shipwreck near Sutivan on island Brač, Croatia. Journal of Archaeological Science. 2015, 62, 143-152.

How to utilize these cultural heritage to generate new value is a current research hotspot [34-35].

34.Lucia, A.; Elisabetta, S.; Marco, C.; Matteo, D.; Matteo, F.; Antonio, I.I.; Roberto, S. Innovative uses of 3D digital technologies to assist the restoration of a fragmented terracotta statue. Journal of Cultural Heritage. 2013, 14, 332-345.

35.Davide, D.; Marco, P.; Andreas, P. The contribution of 3D visual technology to the study of Palaeolithic knapped stones based on refitting. Digital Applications in Archaeology and Cultural Heritage. 2017, 4, 28-38.
